# Sensor Fusion of GNSS and IMU Data for Robust Localization via Smoothed Error State Kalman Filter

**DOI:** 10.3390/s23073676

**Published:** 2023-04-01

**Authors:** Yuming Yin, Jinhong Zhang, Mengqi Guo, Xiaobin Ning, Yuan Wang, Jianshan Lu

**Affiliations:** 1College of Mechanical Engineering, Zhejiang University of Technology, Hangzhou 310014, China; 2College of Engineering, Beijing Forestry University, Beijing 100083, China

**Keywords:** error state Kalman, RTS smoothing, robust and accurate localization

## Abstract

High−precision and robust localization is critical for intelligent vehicle and transportation systems, while the sensor signal loss or variance could dramatically affect the localization performance. The vehicle localization problem in an environment with Global Navigation Satellite System (GNSS) signal errors is investigated in this study. The error state Kalman filtering (ESKF) and Rauch–Tung–Striebel (RTS) smoother are integrated using the data from Inertial Measurement Unit (IMU) and GNSS sensors. A segmented RTS smoothing algorithm is proposed in order to estimate the error state, which is typically close to zero and mostly linear, which allows more accurate linearization and improved state estimation accuracy. The proposed algorithm is evaluated using simulated GNSS signals with and without signal errors. The simulation results demonstrate its superior accuracy and stability for state estimation. The designed ESKF algorithm yielded an approximate 3% improvement in long straight line and turning scenarios compared to classical EKF algorithm. Additionally, the ESKF−RTS algorithm exhibited a 10% increase in the localization accuracy compared to the ESKF algorithm. In the double turning scenarios, the ESKF algorithm resulted in an improvement of about 50% in comparison to the EKF algorithm, while the ESKF−RTS algorithm improved by about 50% compared to the ESKF algorithm. These results indicated that the proposed ESKF−RTS algorithm is more robust and provides more accurate localization.

## 1. Introduction

The Global Navigation Satellite System (GNSS) has found widespread usage in vehicle measurement, intelligent driving, and robot navigation due to its stable long−term localization performance [1,2,3]. However, the susceptibility of satellite signals to external interference would lead to the GNSS having unreliable localization accuracy. Therefore, the combination of GNSS and Inertial Navigation System (INS) has been widely employed, in which both systems could compensate for each other’s drawbacks and leverage their respective strengths to achieve continuous localization [4,5,6]. The reported studies have investigated the combined filtering algorithm of GNSS and INS, GNSS/IMU can provide position, velocity, and attitude information for vehicle control. Liu [7,8] presents a novel pitch and roll feedback mechanism that utilizes intrinsic information of the vehicle such as steering angle and wheel speed. To compensate for the cumulative velocity errors that occur during low sampling intervals of GNSS, the integration of reverse smoothing and grey prediction is employed. Shin and Naser [9] reported that the algorithm yielded better navigation and localization results, even in cases of short−term GNSS loss. Han [10] compared the combined satellite/inertial guidance parameter estimation results with those of a single satellite system, with the former yielding superior results. Li and Zhang [11] studied the combined GNSS/INS navigation algorithm and highlighted that the filtering algorithm still struggled to provide accurate parameter estimation in situations where few satellite data were available. Erfianti [12] researched a combined GNSS/INS navigation algorithm and discovered that the INS could only provide high−accuracy parameter estimation results for a short duration. During the satellite signal occlusion, the forward filtering algorithm alone thus cannot guarantee a robust localization.

Applying the extended Kalman filter (EKF) to estimate the motion of vehicle systems is well desirable due to the system nonlinearity [13,14,15,16]. The EKF linearizes the nonlinear model by approximating it with a first−order Taylor series around the state estimate and then estimates the state using the Kalman filter. M. M. Atia [17] proposed utilizing the extended Kalman filter to merge data from inertial sensors with GNSS data in a loosely coupled mode to improve the accuracy of road network maps in determining lanes. The resulting lane determination success rate was an impressive 97.14%. However, the local linearization operation of this approach can introduce significant estimation errors. Julier [18] introduced an Unscented Kalman filter algorithm that improves the estimation results, while its high complexity makes it unsuitable for most inertial guidance devices. Arasaratnam [19] et al. pointed out through theoretical analysis and simulation verification that the UKF has poor filtering performance or even divergence in solving the high−dimensional (≥20) nonlinear state estimation problem. For this reason, they used the spherical radial rule to approximate the state posterior distribution in the optimal framework, and then proposed the Cubature Kalman Filter (CKF), but the CKF has higher computational complexity and requires more sampling and operations, resulting in poor real−time performance, which is not suitable for some applications with high real−time performance requirements. The extended Kalman filter thus remains the mainstream state estimation algorithm, and developing a low−complexity filter with high accuracy is still challenging [20,21].

To address this challenge, S. S Kourabbaslou [22] presents a flexible design framework utilizing symbolic engines to represent and linearize system and measurement models. A robust fixed−lag smoothing approach is proposed in case there is a mismatch between the nominal model and the actual model [23,24]. To improve the accuracy of vehicle stand−alone localization in highly dynamic driving conditions during GNSS outages, Gao [25] proposed a vehicle localization system based on vehicle chassis sensors considering vehicle lateral velocity. The Kalman filter combines vehicle states obtained from vehicle kinematics and dynamics to improve the reliability and accuracy of autonomous driving. A consensus−based and vehicle kinematics/dynamics integrated autonomous vehicle sideslip angle estimation algorithm based on GNSS/INS was proposed [26,27]. Madyastha [28] proposed a Kalman filtering method based on attitude error states. The error state Kalman filter (ESKF) is designed for covariance estimation, which serves as a weight for the inertial odometry optimization process [29,30,31]. These results validate the ESKF’s effectiveness for error−state estimation, although a full−state estimation based on ESKF is not available yet. Further exploration of state estimation based on ESKF is thus favorable.

The major challenge in designing a sensor fusion algorithm for state estimation is to address GNSS signal errors and low−cost hardware limitations with a smoother approach. The errors of inertial navigation systems accumulate over time, while GNSS signals are heavily influenced by satellite signal quality and urban obstructions [32,33,34]. These would lead to significant localization errors. Using INS alone cannot provide accurate position estimation for a long term in signal−obscured environments [35,36,37]. In such cases, real−time high−accuracy localization is necessary, and data fusion from multiple sensors is often required [38,39]. In order to provide reliable localization information, simultaneous localization and map construction (SLAM) systems can be used, as well as multi−sensor integrated systems that combine LiDAR [40] and navigation systems to solve problems caused by GNSS antenna failures. To reduce GNSS and Inertial Measurement Unit (IMU) fusion costs and achieve desired accuracy, different navigation satellite [41,42,43] constellations need to be combined, especially in urban working environments.

In this paper, the low−cost sensor combination of only GNSS and IMU is considered, and effects of GNSS signal error and different fusion algorithm design are investigated. The extended Kalman filter and error state Kalman filter are designed in Section 2. The application of the error−extended Kalman filter and the RTS optimal smoothing algorithm is investigated in Section 3 for combined inertial guidance and GNSS localization in urban working environments. The insights and conclusions on the performance of the designed sensor fusion algorithms are further analyzed.

## 2. GNSS and IMU Integrated Filter Design

The overall sensor fusion framework integrating the GNSS and IMU sensor data with significant GNSS signal errors is illustrated in Figure 1. It mainly consists of four procedures, including data analysis, prediction process, update process and reverse smoothing, contributing to the developed ESKF−RTS smoothing localization algorithm. In particular, the algorithm defines a nominal state without considering the measurement noise of IMU and system disturbance, and an error state containing the noise and disturbance information is used for state estimation. The nominal state condition and error state prediction are updated simultaneously, and the error state is corrected using GNSS signal measurement and injected into the nominal state. The error state and its covariance matrix are subsequently reset. The acquisition frequency for GNSS data is 1 Hz, while the IMU data are acquired at a frequency of 100 Hz. The reverse time update and reverse segmentation smoothing are parallelly performed, which will be detailed below.

### 2.1. Extended Kalman Filter

The extended Kalman filter (EKF) is also designed for comparison, using the sensor measurements directly. The EKF procedure is described briefly.

The developed discrete nonlinear vehicle system equations are:(1)$$\left\{ \begin{array}{l} X_{k}=f(X_{k-1})+W_{k-1}\\ Z_{K}=h(X_{k})+V_{k} \end{array}\right.,$$
where f(·) and h(·) represent the state function and measurement function of nonlinear vehicle systems, W and V are Gaussian white noise of state X and measurement Z, respectively, and k represents the discrete time step.

The state function of the nonlinear vehicle system in Equation (1) is approximated through local linearization. The nonlinear equation is expanded using a Taylor series, and only the first−order term is retained while ignoring the second and higher−order terms. This yields the following expression:(2)$$X_{k}=f({\overset{⌣}{X}}_{k-1})+F_{k-1}(X_{k-1}-{\overset{⌣}{X}}_{k-1})+W_{k-1},$$
where $F_{k-1}$ is the Jacobian matrix of the function of the function $f(X_{k-1})$.

The statistical characteristics of the predicted state can be obtained based on the results of the linearization in the form of
(3)$${\overset{⌢}{X}}_{k,k-1}=E[f({\overset{⌣}{X}}_{k-1})+F_{k-1}(X_{k-1}-{\overset{⌣}{X}}_{k-1})+W_{k-1}]=f({\overset{⌣}{X}}_{k-1}).$$

It is possible to further predict the variance matrix as
(4)$$\begin{array}{ll} P_{k,k-1} & =E[(X_{k}-{\overset{⌣}{X}}_{k,k-1}){\left(X_{k}-{\overset{⌣}{X}}_{k,k-1}\right)}^{T}]\\ & =E\{[F_{k-1}(X_{k-1}-{\overset{⌣}{X}}_{k-1})+W_{k-1}]{\left[F_{k-1}(X_{k-1}-{\overset{⌣}{X}}_{k-1})+W_{k-1}\right]}^{T}\}.\\ & =F_{k-1}P_{k-1}F_{{}_{k-1}}^{T}+Q_{k-1} \end{array}$$

The same method is used for the measurement function $h\left(x_{k}\right)$ at the point ${\hat{x}}_{k}\;$, which can be obtained as
(5)$$Z_{K}=h({\overset{⌣}{X}}_{k})+H_{k}(X_{k}-{\overset{⌣}{X}}_{k})+V_{k},$$
where $H_{k}$ is the Jacobian matrix of the function of the function $h(X_{k-1})$. The prediction of measurement is calculated as
(6)$${\overset{⌢}{Z}}_{k,k-1}=E[h({\overset{⌣}{X}}_{k-1})+H_{k-1}(X_{k-1}-{\overset{⌣}{X}}_{k-1})+V_{k-1}]=h({\overset{⌣}{X}}_{k-1}).$$

Then, the collaborative variance can be further predicted as
(7)$$\begin{array}{ll} P_{k,k-1} & =E[(Z_{k}-{\overset{⌣}{Z}}_{k,k-1}){\left(Z_{k}-{\overset{⌣}{Z}}_{k,k-1}\right)}^{T}]\\ & =E\{[H_{k-1}(X_{k-1}-{\overset{⌣}{X}}_{k-1})+V_{k-1}]{\left[H_{k-1}(X_{k-1}-{\overset{⌣}{X}}_{k-1})+V_{k-1}\right]}^{T}\}.\\ & =H_{k-1}P_{k-1}H_{k-1}^{T}+R_{k-1} \end{array}$$

Using the information above, the iterative process of the EKF algorithm for the sensor fused localization can be summarized as
(8)$$\begin{array}{l} {\overset{⌣}{X}}_{k}=f({\overset{⌣}{X}}_{k-1})\\ {\overset{⌣}{P}}_{k}=F_{k-1}P_{k-1}F_{{}_{k-1}}^{T}+Q_{k-1}\\ K_{k+1}={\overset{⌣}{P}}_{k+1}^{-}H_{k+1}^{T}{\left(H_{k+1}{\overset{⌣}{P}}_{k+1}^{-}H_{k+1}^{T}+V_{k+1}\right)}^{-1}.\\ X_{k+1}=K_{k+1}(Z_{k+1}-h({\overset{⌣}{X}}_{k+1}))\\ P_{k+1}=(I-K_{k+1}H_{k+1}){\overset{⌣}{P}}_{k+1}^{-} \end{array}$$

### 2.2. Error State Kalman filter

For the formulation of the error state filter, the system states are defined as the true, nominal, and error state values. The true state is expressed as a combination of the nominal and error states. The approach is to treat the nominal state as a dominant signal, which is highly nonlinear, and the error state as a small signal, which is linearly effective and suitable for linear Gaussian filtering. The nominal state vector of ${\left[p,v,q,a_{b,}\omega_{b}\right]}^{T}$ is used in this study, where the additional dimension appears due to the quaternion representation used for rotation, the quaternion method is widely used in attitude update because it only requires the calculation of matrix differential equations, which is relatively small in computation and can avoid the singular value problem of Euler angles. In this paper, the quaternion method is used to solve the attitude of the carrier. The operators q and R represent, respectively, the quaternion corresponding to the axial angular vector θ and its rotation matrix. In addition, the error state vector is ${\left[\delta p,\delta v,\delta q,\delta a_{b,}\delta \omega_{b}\right]}^{T}$. The relevant symbol definitions of true, nominal, and error state are listed and compared in Table 1.

#### 2.2.1. Continuous Time Kinetic Model

The nominal state kinematics refers to the system modelled with system noise or external perturbations in the form of
(9)$$\begin{array}{l} {\dot{p}}_{t}=v_{t}\\ {\dot{v}}_{t}=R_{t}(a_{m}-a_{bt}-a_{n})+g_{t}\\ {\dot{q}}_{t}=\frac{1}{2}q_{t}⊗(\omega_{m}-\omega_{bt}-\omega_{n}).\\ {\dot{a}}_{bt}=a_{\omega }\\ {\dot{\omega }}_{bt}=\omega_{\omega } \end{array}$$

The acceleration and angular velocity measurements are represented by $a_{m}$ and $\omega_{m}$, respectively, while $a_{n}$ and $\omega_{n}$ represent the corresponding noise vectors, and $a_{\omega }$ and $\omega_{\omega }$ represent the bias vectors for acceleration and angular velocity, respectively. The equations driving the system dynamics in the nominal state are as follows, where the nominal state refers to the modeled system without any noise or perturbations:(10)$$\begin{array}{l} \dot{p}=v\\ \dot{v}=R(a_{m}-a_{b})+g\\ \dot{q}=\frac{1}{2}q⊗(\omega_{m}-\omega_{b}).\\ {\dot{a}}_{b}=0\\ {\dot{\omega }}_{b}=0 \end{array}$$

Solving for the error state and simplifying all second−order infinitesimals. The kinetic equation for the error state can be obtained from Equations (9) and (10).
(11)$$\begin{array}{l} \delta \dot{p}=\delta v\\ \delta \dot{v}=R(a_{m}-a_{b})\delta \theta -R\delta a_{b}-Ra_{n}\\ \delta \dot{\theta }=-(\omega_{m}-\omega_{b})\delta \theta -\delta \omega_{b}-\omega_{n}.\\ \delta {\dot{a}}_{b}=a_{\omega }\\ \delta {\dot{\omega }}_{b}=\omega_{\omega } \end{array}$$

#### 2.2.2. Discrete Time Kinetic Model

In order to apply the derived state and measurement equations to the sensor fusion filter, they must be discretized based on the sampling time. The original continuous time equations are transformed into their discrete time form.

The recursive expression for the motion model of the nominal state is
(12)$$\begin{array}{l} p_{\left(k+1\right)}=p_{k}+v_{k}\Delta t+\frac{1}{2}[R_{k}(a_{mk}-a_{bk})+g]\Delta t^{2}\\ v_{\left(k+1\right)}=v_{k}+[R_{k}(a_{mk}-a_{bk})+g_{k}]\Delta t\\ q_{\left(k+1\right)}=q_{k}⊗q_{k}((\omega_{mk}-\omega_{bk})\Delta t)\\ a_{b}(k+1)=a_{bk}\\ \omega_{b}(k+1)=\omega_{bk} \end{array}$$

The kinematic model of the error state is expressed recursively as a function of the non−negative integer *k* representing the kth time step:(13)$$\begin{array}{l} \delta p_{k+1}=\delta p_{k}+\delta v_{k}\Delta t\\ \delta v_{k+1}=\left[-R_{k}(a_{mk}-a_{bk})\delta \theta_{k}-R_{k}\delta a_{bk}\right]\Delta t+\delta v_{k}-w_{vk}\\ \delta \theta_{k+1}=R_{k}^{T}\left((\omega_{mk}-\omega_{bk})\Delta t\right)\delta \theta_{k}-\delta \omega_{bk}\Delta t+w_{\theta k}\\ \delta a_{b(k+1)}=\delta a_{bk}+w_{ak}\\ \delta \omega_{b(k+1)}=\delta \omega_{bk}+w_{wk} \end{array}$$

The Gauss random noise for velocity, attitude, acceleration bias, and angular velocity bias are denoted as $w_{vk},w_{\theta k},w_{ak},,w_{\omega k}$, their mean is zero, and their covariance matrices are obtained by integrating the covariances of $\sigma_{a_{n}}$, $\sigma_{\omega_{n}}$, $\sigma_{a_{\omega }}$ and $\sigma_{\omega_{\omega }}$ over the step time Δt.
(14)$$\begin{array}{l} W_{v}=\sigma_{a_{n}}^{2}\Delta t^{2}I\\ W_{\theta }=\sigma_{\omega_{n}}^{2}\Delta t^{2}I\\ W_{A}=\sigma_{a_{\omega }}^{2}\Delta tI\\ W_{\Omega }=\sigma_{\omega_{\omega }\Delta t}^{2}I \end{array}$$

The standard deviation of Gaussian white noise for acceleration and angular velocity are denoted by $\sigma_{a_{n}}$ and $\sigma_{\omega_{n}}$, respectively. Similarly, $\sigma_{a_{\omega }}$ and $\sigma_{\omega_{\omega }}$ are used to represent the standard deviation of Gaussian white noise for acceleration and angular velocity bias, respectively.

#### 2.2.3. Development of the Error State Model

The discrete vector forms for all states, error states, IMU measurements, and noise are defined as follows:(15)$$\begin{array}{l} x_{k}={\left[x_{k},v_{k},q_{k},a_{bk},\omega_{bk}\right]}^{T},\delta x_{k}={\left[\delta p_{k},\delta v_{k},\delta q_{k},\delta a_{bk},\delta \omega_{bk}\right]}^{T},\\ u_{mk}={\left[a_{mk},\omega_{mk}\right]}^{T},w_{k}={\left[w_{vk},w_{\theta k},w_{ak},w_{\omega k}\right]}^{T} \end{array}$$

The recursive equation for all the error state is derived by combining Equations (5) and (7) in the form of
(16)$$\delta x_{k+1}=f_{\delta }(x_{k},\delta x_{k},u_{mk},w_{k})=F_{xk}(x_{k},u_{mk})+F_{wk}w_{k},$$
where $f_{\delta }\left(\cdot \right)$ represents the recursive function of the error state, the $F_{xk}$ and $F_{wk}$ are the Jacobian matrices corresponding to the error and noise states, respectively, which can be derived as
(17)$$F_{xk}=\left[\begin{array}{ccccc} I & I\Delta t & 0 & 0 & 0\\ 0 & I & -R_{k}^{T}(\omega_{mk}-\omega_{bk})\Delta t & -R_{k}\Delta t & 0\\ 0 & 0 & R_{k}^{T}(\omega_{mk}-\omega_{bk})\Delta t & 0 & -I\Delta t\\ 0 & 0 & 0 & I & 0\\ 0 & 0 & 0 & 0 & I \end{array}\right],F_{wk}=\left[\begin{array}{cccc} 0 & 0 & 0 & 0\\ I & 0 & 0 & 0\\ 0 & I & 0 & 0\\ 0 & 0 & I & 0\\ 0 & 0 & 0 & I \end{array}\right].$$

#### 2.2.4. The ESKF Prediction Process

The error states and the covariance prediction process can be obtained as
(18)$$\begin{array}{l} \delta {\overset{⌣}{x}}_{k+1}^{-}=F_{xk}({\overset{⌢}{x}}_{k},u_{mk})\delta x_{k}\\ {\overset{⌣}{P}}_{\left(k+1\right)}^{-}=F_{xk}{\overset{⌣}{P}}_{k}F_{xk}^{T}+F_{wk}Q_{w}F_{wk}^{T} \end{array}$$
where $Q_{w}$ represents the noise covariance matrix of the form:(19)$$Q_{w}=\left[\begin{array}{cccc} W_{v} & 0 & 0 & 0\\ 0 & W_{\theta } & 0 & 0\\ 0 & 0 & W_{A} & 0\\ 0 & 0 & 0 & W_{\Omega } \end{array}\right].$$

#### 2.2.5. The ESKF Observation Process

Once the GNSS information becomes available, the observation is incorporated to continuously update the filter. This process also involves calibrating for the accelerometer and gyroscope biases.

The observation equation is typically expressed in a more uniform form within the filter in the form of
(20)$$z_{k}=h(x_{tk})+w_{mk},$$
where $z_{k}$ is the measurement signal vector, $w_{mk}$ represents the Gaussian white noise of the measurement signal, and its covariance is V. The error calibration update equation is derived as
(21)$$\begin{array}{l} K_{k+1}={\overset{⌣}{P}}_{k+1}^{-}H_{k+1}^{T}{\left(H_{k+1}{\overset{⌣}{P}}_{k+1}^{-}H_{k+1}^{T}+V_{k+1}\right)}^{-1}\\ \delta {\overset{⌢}{X}}_{k+1}=K_{k+1}(Z_{k+1}-H({\overset{⌣}{X}}_{k+1}))\\ {\overset{⌢}{P}}_{k+1}=(I-K_{k+1}H_{k+1}){\overset{⌣}{P}}_{k+1}^{-} \end{array}$$

The Jacobian matrix ***H*** is required to be defined with respect to the error state δx, and evaluated at the best true state estimate ${\hat{x}}_{t}=x\;⊕\;\delta \hat{x}$. As the error state mean is zero at this stage (we have not observed it yet), we have $x_{t}=x,$ and we can use the nominal error x as the evaluation point, leading to
(22)$$H={\frac{\partial h}{\partial x_{t}}|}_{x}{\frac{\partial x_{t}}{\partial \delta x}|}_{x}=H_{x}X_{\delta x},$$
in which
(23)$$X_{\delta x}≜{\frac{\partial x_{t}}{\partial \delta x}|}_{x}=\left[\begin{array}{ccc} I_{6} & 0 & 0\\ 0 & Q_{\delta \theta } & 0\\ 0 & 0 & I_{6} \end{array}\right],$$
(24)$$Q_{\delta \theta }=\frac{1}{2}\left[\begin{array}{ccc} -q_{x} & -q_{y} & -q_{z}\\ q_{w} & -q_{z} & q_{y}\\ q_{z} & q_{w} & -q_{x}\\ -q_{y} & q_{x} & q_{w} \end{array}\right].$$

#### 2.2.6. Combination of ESKF Error State and Nominal State Process

Based on the recurrence of the a priori system state and the calibration of the error state, the updated form is obtained as follows:(25)$${\overset{⌢}{x}}_{k}={\overset{⌢}{x}}_{{}_{k}}^{-}⊕\delta {\overset{⌢}{x}}_{k}.$$

Each of these system states corresponds to the following:(26)$$\begin{array}{l} {\overset{⌢}{p}}_{k}={\overset{⌢}{p}}_{k}^{-}+\delta {\overset{⌢}{p}}_{k}\\ {\overset{⌢}{v}}_{k}={\overset{⌢}{v}}_{k}^{-}+\delta {\overset{⌢}{v}}_{k}\\ {\overset{⌢}{q}}_{k}={\overset{⌢}{q}}_{k}^{-}⊗\overset{⌢}{q}\delta {\overset{⌢}{\theta }}_{k}.\\ {\overset{⌢}{a}}_{bk}={\overset{⌢}{a}}_{bk}^{-}+\delta {\overset{⌢}{a}}_{bk}\\ {\overset{⌢}{\omega }}_{bk}={\overset{⌢}{\omega }}_{bk}^{-}+\delta {\overset{⌢}{\omega }}_{bk} \end{array}$$

#### 2.2.7. ESKF Error State Reset

After injecting the error state into the nominal state, the a priori error state and the corresponding covariance need to be reset, as
(27)$$\begin{array}{l} \delta \overset{⌢}{x}=0\\ P=G_{k}P_{k}G_{k}^{T} \end{array}$$
where G is the reset function $g\left(\delta x\right)=\delta x⊖\delta \hat{x}$ of the Jacobi matrix, which is defined in the form of
(28)$$G≜{\frac{\partial g}{\partial \delta x}|}_{\delta \overset{⌢}{x}}=\left(\begin{array}{ccc} I_{6} & 0 & 0\\ 0 & I-(\frac{1}{2}\delta \overset{⌢}{\theta }) & 0\\ 0 & 0 & I_{9} \end{array}\right).$$

## 3. Robust Localization via RTS Smoothing

### 3.1. Reliability of Measurement Information

In the case of a GNSS signal loss, when there are no available GNSS measurements to update, the covariance matrix in Equation (21) tends to be infinite. The Kalman gain K tends to zero, so that when the measurement information is zero or differs too much from the predicted information, the update equation changes to the following form:(29)$$\begin{array}{l} \delta {\overset{⌢}{x}}_{k+1}=\delta {\overset{⌢}{x}}_{k}\\ {\overset{⌢}{P}}_{k+1}={\overset{⌣}{P}}_{k}^{-} \end{array}$$

### 3.2. RTS Fundamental Design

The RTS smoothing can be regarded as a technique for obtaining an optimal state estimate when observations are available from moment 1 to moment *N*; it involves using previous estimates obtained through Kalman filtering in order to perform backward smoothing from moment *k +* 1 to moment *k* resulting in a more precise estimate. This method falls under the category of fixed interval smoothing and is considered as a fixed interval smoother, which is convenient for implementation.

The RTS smoother involves a two−step process: a forward filter followed by a backward smoothing. The forward filter is a standard Kalman filter described by Equation (18), which maintains all the predicted and updated estimates as well as their corresponding covariances for each epoch during the entire mission. The backward smoothing procedure begins at the end of the forward filter at time $t_{N}$, with an initial condition $\delta x_{N,N}$ and $t_{N,N}$. Therefore, the backward smoothing can be seen as an update to the forward filter for obtaining an enhanced estimate. The equation for the RTS algorithm can be represented in the form of
(30)$$\begin{array}{l} \delta {\overset{⌢}{x}}_{k,N}=\delta {\overset{⌢}{x}}_{k,k}+U_{k}({\overset{⌢}{x}}_{k+1,N}-{\overset{⌢}{x}}_{k+1,k})\\ {\overset{⌢}{P}}_{k,N}={\overset{⌢}{P}}_{k,k}+U_{k}({\overset{⌢}{P}}_{k+1,N}-{\overset{⌢}{P}}_{k+1,k})U_{k}^{T} \end{array}$$
where delta $\delta x_{k,N}$ is the smoothed estimate of the state vector, and $U_{k}$ is the smoothing gain matrix, which is calculated by the following equation:(31)$$U_{k}={\overset{⌢}{P}}_{k,k}F_{k+1,k}^{T}{\overset{⌢}{P}}_{k+1,k}^{-1},$$
where k=N−1,N−2,……,0. The RTS smoothing algorithm uses the results of the forward filter and backward smoothing to obtain an improved estimate. The inverse smoothing starts from the last filter result and proceeds forward one by one, so it must obtain N filter results to smooth the observations within the period [0, N]. However, if N is too large, the inverse smoothing process can be obviously lagged, which would limit the algorithm’s application in real time. Therefore, a segmented RTS smoothing method is designed in this study.

### 3.3. Segmented RTS Smoothing Algorithm

Assuming the duration of a dynamic system is *k* = 1, 2……, *N*, the segmented RTS smoothing algorithm involves forward filtering and inverse smoothing of N observations in segments. The segment length is *L*, where 1 < *L* < *N*. In other words, the RTS inverse smoothing process is performed immediately after obtaining the filtering results for the segmented period, without waiting for subsequent filtering results. This approach significantly reduces the lag time of the inverse smoothing process. Additionally, the segmentation length can be varied, avoiding the issue of poor real−time performance associated with RTS smoothing. In this study, the segmented RTS smoothing algorithm is applied to the potential probability assumption density filtering algorithm, effectively addressing the problem of poor real−time performance resulting from the use of RTS smoothing.

Step 1 Output variables are cleared: $\delta {\hat{x}}_{k}=\left[\;\right]$,$\;{\hat{p}}_{k}=\left[\;\right]$.

Step 2 Local filtering result variable clearing $\delta {\overset{˘}{x}}_{k}=\left[\;\right]$,${\overset{˘}{p}}_{k}=\left[\;\right]$. The filtered result variable after local association is cleared $x_{k}=\left[\;\right]$,$\;Q_{k}=\left[\;\right]$, the segmented smoothing counter count is cleared to zero.

Step 3 If the tracking of the target is not finished, action continues down the track, otherwise the smoothing result $\delta {\hat{x}}_{k}$,$\;{\hat{p}}_{k}$ is output; then, the entire segmented RTS smoothing algorithm ends.

Step 4 Upon receiving the current filtering result, it is stored in the variables $\delta {\overset{˘}{x}}_{k}$ and ${\overset{˘}{p}}_{k}$. the Hungarian algorithm is used to correlate the trajectories and estimates, obtaining the filter value for each target. The correlated filter result is saved in variables $x_{k}$ and $Q_{k}$, and they are counted using a counter called count.

Step 5 If count==L, after performing local correlation using $x_{k}$ and $Q_{k}$ for each target’s filtering results, the inverse smoothing process uses the RTS smoothing algorithm. The results of the smoothing process are stored in variables delta $\delta {\overset{˘}{x}}_{k}$ and ${\overset{˘}{p}}_{k}$ before being transferred to Step 2. In order to significantly reduce the lag time of the inverse smoothing process, it is carried out immediately following the acquisition of the filtered data without waiting for subsequent data. Real−time performance of the fusion algorithm can be improved by decreasing the segment length L.

To correct the position, velocity, and attitude values that have been calculated in the forward filtering, it is recommended to use the state error smoothing values. These smoothing values can be utilized to derive the final optimal smoothed values for these states using the following equations.

## 4. Results and Discussions

The proposed sensor fusion algorithm is demonstrated in a relatively open environment, which allows for uninterrupted satellite signal and individualized GNSS localization. The acquisition frequency for GNSS data is 1 Hz, while the IMU data are acquired at a frequency of 100 Hz; the smooth dimension L is selected as 10. These parameters provide acceptable conditions for analyzing the accuracy of combined GNSS/IMU localization, even in the event of GNSS signal loss, as demonstrated by the trajectory shown in Figure 2. In order to evaluate the localization accuracy of the GNSS/IMU combined localization process, the simulation involves artificially inducing the loss of lock by introducing exaggerated errors to the GNSS satellite observation information based on raw GNSS data.

### 4.1. Oval Track Simulation Analysis

Figure 2 illustrates the simulated oval shape trajectory, which comprises a long straight line with small angle turns in two dimensions.

The localization results for the various filtering algorithms are shown in Figure 3, Figure 4 and Figure 5 and Table 2. It can be seen that the ESKF algorithm exhibits better target tracking accuracy compared to the EKF algorithm, as evidenced by its RMS values of 1.633 m, 1.782 m, and 1.476 m for Lateral, Longitudinal, and Vertical directions, respectively. By reducing the error state parameter, the accuracy of the three poses is improved by 2.8%, 2.1%, and 52.0%, respectively. After performing the backward smoothing filtering process, the RMS values of the three attitudes are further optimized to 1.463 m, 1.588 m, and 1.393 m, respectively. Moreover, the target tracking accuracy of ESKF–RTS is superior to ESKF, with the attitude accuracy in the three directions improving by 10.4%, 10.9%, and 5.6%, respectively. The trajectory also reveals that the curve of ESKF–RTS is smoother, further highlighting the advantages of ESKF–RTS over the other two algorithms in addressing the problem of smooth estimation of nonlinear states.

### 4.2. Serpentine Track Simulation Analysis

Further verification of the designed algorithms is performed in a serpentine trajectory, as shown in Figure 6, and the localization results are compared as shown in Figure 7, Figure 8 and Figure 9 and Table 3.

Table 3 illustrates the improved performance of the ESKF and ESKF−RTS algorithms compared to the EKF algorithm in the serpentine condition. The RMS of Lateral, Longitudinal, and Vertical in EKF are 0.955 m, 1.585 m, and 5.823 m, respectively. After reducing the state value, the RMS of the three positions for the ESKF algorithm are 0.464 m, 0.641 m, and 1.700 m, which are 51.4%, 59.6%, and 70.8% better than the EKF algorithm. The ESKF−RTS algorithm further improves the position accuracy, with RMS values of 0.206 m, 0.243 m, and 0.912 m for the three directions, respectively, which are 55.6%, 62.1%, and 46.4% higher than those of the ESKF algorithm. Figure 7, Figure 8 and Figure 9 show the comparison of the root mean square errors, which further demonstrates the superior performance and stability of the ESKF−RTS algorithm. However, since the simulation was conducted with a relatively good GNSS signal, the robustness of the algorithm could not be well evaluated. Subsequently, the GNSS data for the serpentine condition are partitioned into eight segments, and the error magnitude is amplified by a factor of two at intervals of 600 s in order to evaluate the algorithm’s robustness.

The serpentine working condition with further exaggerated GNSS signal error is analyzed, as shown in Figure 10, Figure 11 and Figure 12 and Table 4. The RMS values for the Lateral, Longitudinal, and Vertical directions are 1.231 m, 1.735 m, and 1.453 m, respectively, for the EKF algorithm. Meanwhile, the ESKF algorithm improves these values by 48.4%, 48.7%, and 34.1%, respectively, resulting in the RMS values of 0.635 m, 0.890 m, and 0.957 m. Moreover, the ESKF−RTS algorithm improves the position accuracy in three directions by 42.1%, 52.6%, and 52.1%, respectively, compared to the ESKF algorithm, with RMS values of 0.368 m, 0.422 m, and 1.456 m. Even though the accuracy of ESKF−RTS decreases on the Vertical axis, the higher degree of smoothing in Figure 12 suggests that the ESKF−RTS smoothing algorithm can significantly enhance robustness and achieve precise localization, even with lower accuracy of GNSS sensors.

Next, we continued the simulation of circular operating conditions and divided the entire simulation process into four sections to validate the feasibility of the algorithm by doubling the GNSS signal error, as shown in Figure 13, Figure 14, Figure 15 and Figure 16. The experimental results shown in Table 5 once again demonstrated the feasibility and robustness of the error state Kalman filtering algorithm, indicating that the algorithm can achieve stable and accurate integrated navigation under various operating conditions.

The above data show the results of testing the navigation system using EKF, ESKF, and ESKF−RTS algorithms. By analyzing these results, we can draw the following conclusions. The above data demonstrate the performance of three different filtering algorithms in GNSS navigation. First, we can see that the performance of the EKF and ESKF algorithms is relatively similar, with position errors around 1.3 m. This is because both algorithms use Kalman filtering, but the ESKF algorithm introduces error states while considering errors, which can more accurately estimate errors and improve accuracy. At the same time, the ESKF algorithm can also reduce the impact of errors in future time by predicting error states, thus improving the stability of the algorithm.

On the other hand, the ESKF−RTS algorithm performs even better, with position errors even lower than 1 m. This is because the ESKF−RTS algorithm uses segmental smoothing to optimize the filtering results, which can more accurately estimate and correct errors. In the ESKF−RTS algorithm, RTS stands for “Recursive Least Squares Smoothing” which can combine the prediction results of the ESKF algorithm with the observation results to obtain more accurate state estimation results.

Overall, the above data indicate that the ESKF−RTS algorithm performs well in GNSS navigation. The advantage of this algorithm is that it not only considers error states, but also further optimizes the filtering results through smoothing algorithms. Therefore, in practical applications, selecting the ESKF−RTS algorithm for navigation filtering can achieve more accurate and stable results, improving the reliability and accuracy of the navigation system.

## 5. Conclusions and Future Work

This paper introduces an error state extended Kalman filter algorithm and segmented Rauch–Tung–Striebel (RTS) smoothing algorithm to enhance the localization accuracy and robustness of GNSS and IMU sensors. The cumulative error of INS over time are overcome when GNSS signal is disturbed. The simulation results show that the proposed method is more linear and has higher localization accuracy than the traditional EKF algorithm. It also demonstrates good robustness in achieving better accuracy with low−quality GNSS signals. The algorithm can serve as a foundation for low−cost sensor fusion processing and is a valuable reference for further research.

Over the next few years, combined navigation systems will see increased usage and development across a range of applications. As sensor and communication technologies progress, and the demands for navigation safety and reliability continue to rise, there will be a greater need for more accurate and dependable navigation systems.

Furthermore, the RTS smoothing algorithm−assisted combined navigation algorithm presented in this paper was simulated for offline computation. For practical applications in the future, real−world testing will be necessary to verify the hardware feasibility of the algorithm, and to consider real−time data transmission and computation to achieve unmanned driving. Overall, positioning is a crucial component of intelligent driving, and its potential impact on society and the economy is vast. Combined navigation will be more and more widely used in production and practical life.

## Figures and Tables

**Figure 1 sensors-23-03676-f001:**
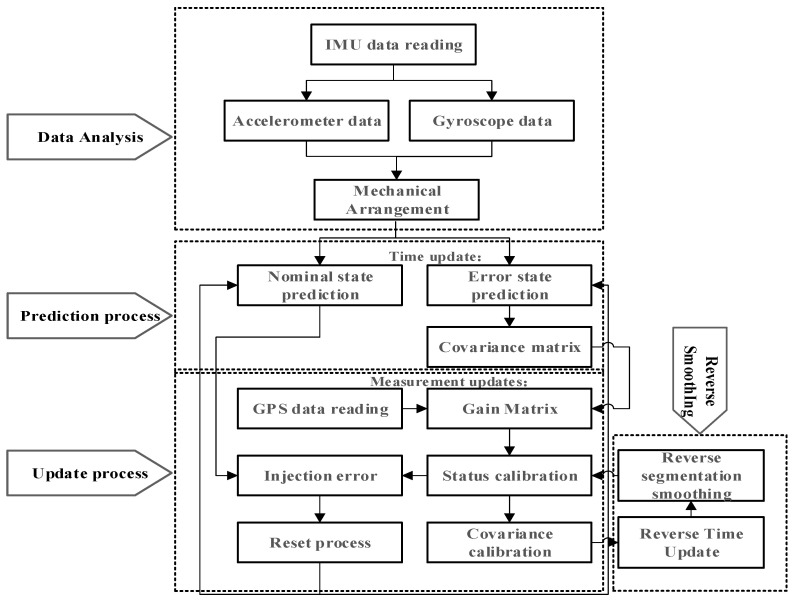
Flow chart of the designed ESKF−RTS algorithm.

**Figure 2 sensors-23-03676-f002:**
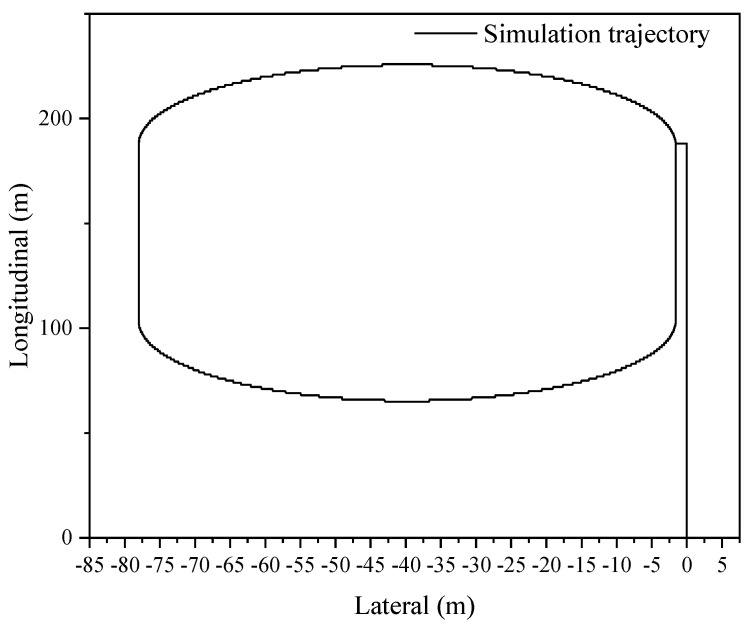
Oval shape motion trajectory.

**Figure 3 sensors-23-03676-f003:**
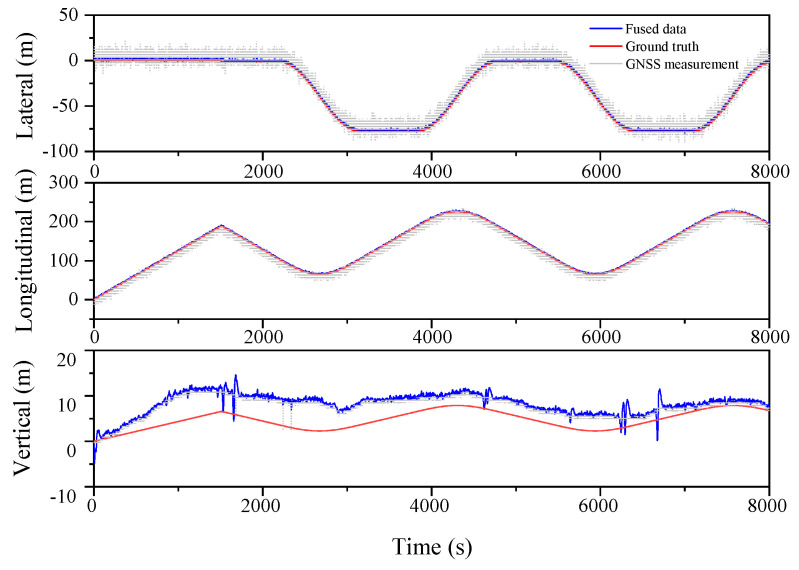
Lateral, Longitudinal and Vertical positions of EKF (Oval).

**Figure 4 sensors-23-03676-f004:**
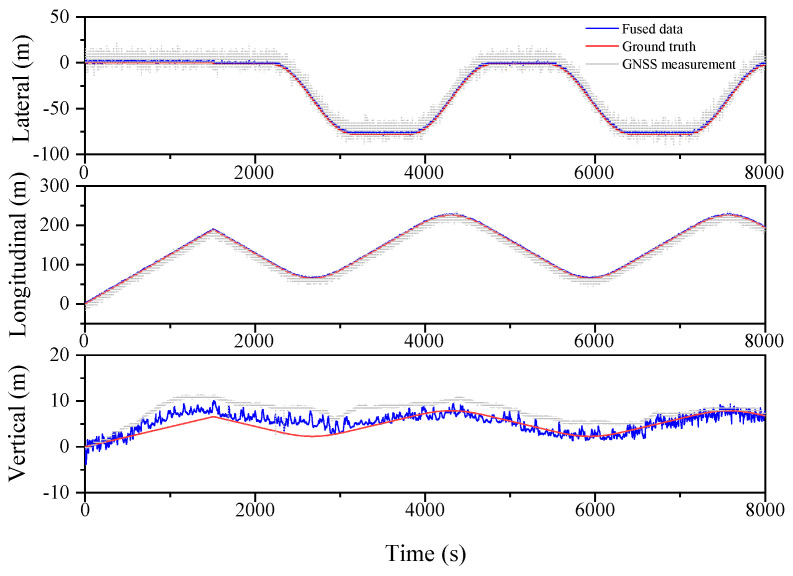
Lateral, Longitudinal and Vertical positions of ESKF (Oval).

**Figure 5 sensors-23-03676-f005:**
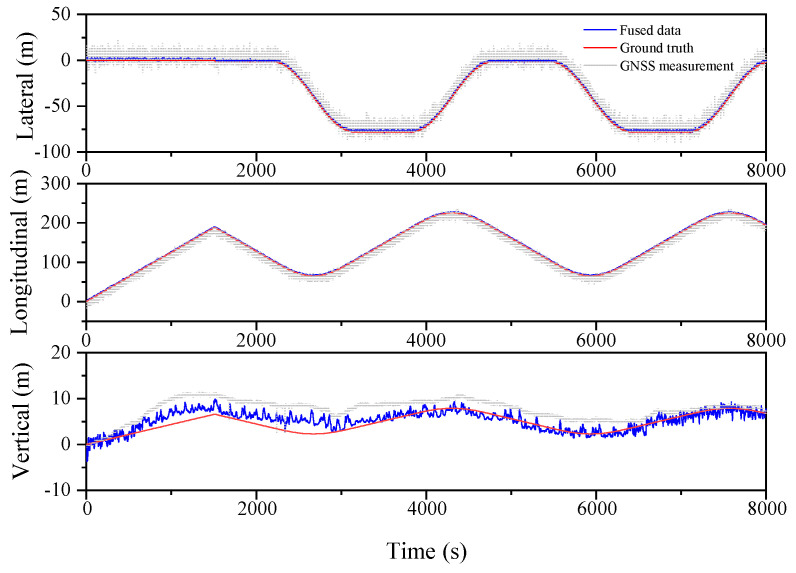
Lateral, Longitudinal and Vertical positions of ESKF–RTS (Oval).

**Figure 6 sensors-23-03676-f006:**
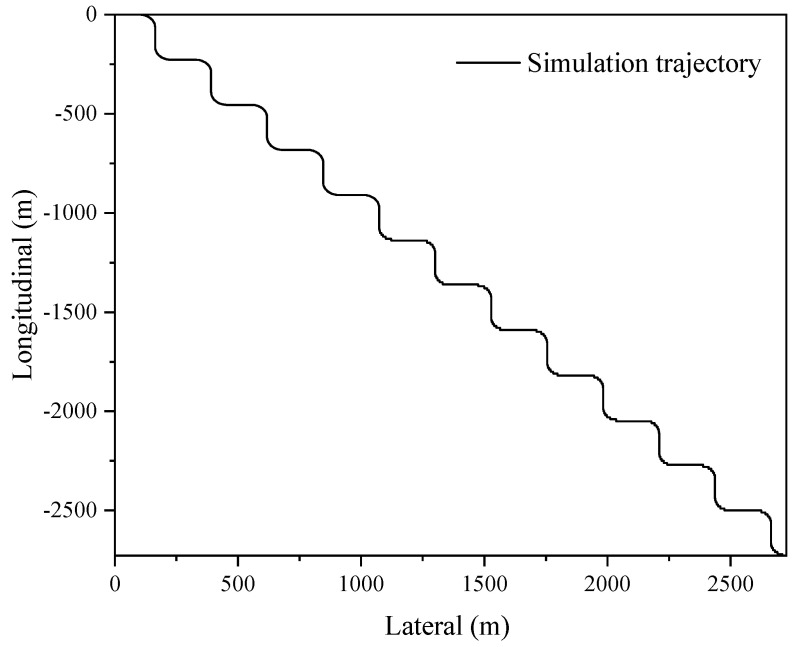
Serpentine shape motion trajectory.

**Figure 7 sensors-23-03676-f007:**
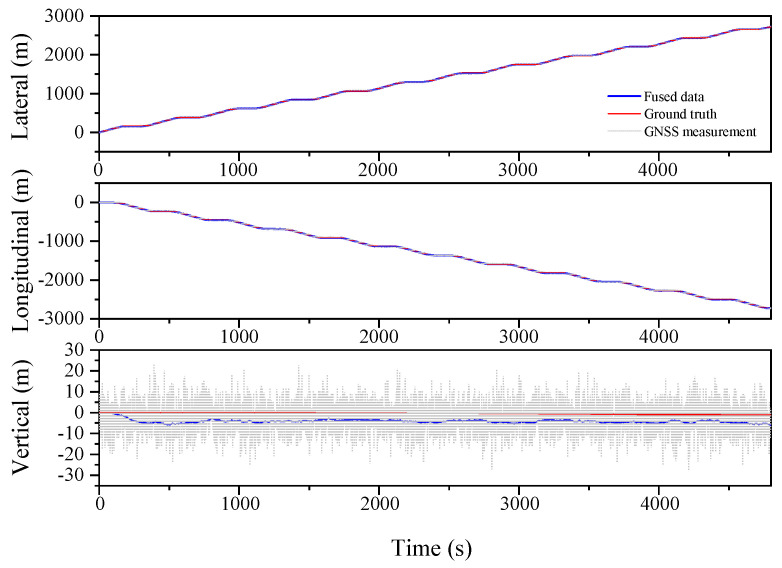
Lateral, Longitudinal and Vertical positions of EKF (Serpentine 1).

**Figure 8 sensors-23-03676-f008:**
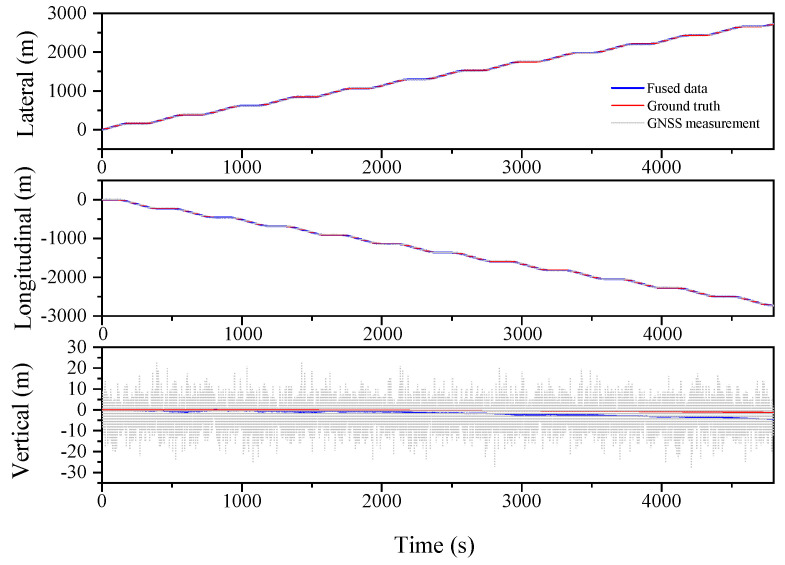
Lateral, Longitudinal and Vertical positions of ESKF (Serpentine 1).

**Figure 9 sensors-23-03676-f009:**
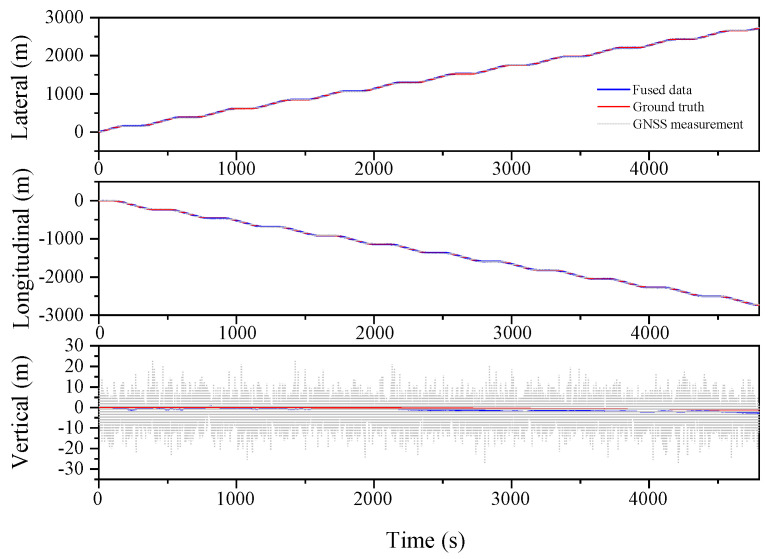
Lateral, Longitudinal and Vertical positions of ESKF−RTS (Serpentine 1).

**Figure 10 sensors-23-03676-f010:**
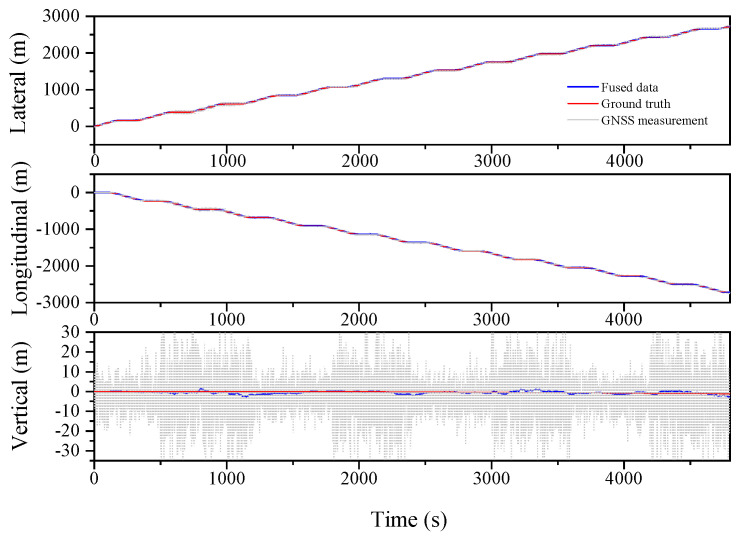
Lateral, Longitudinal and Vertical positions of EKF (Serpentine 2).

**Figure 11 sensors-23-03676-f011:**
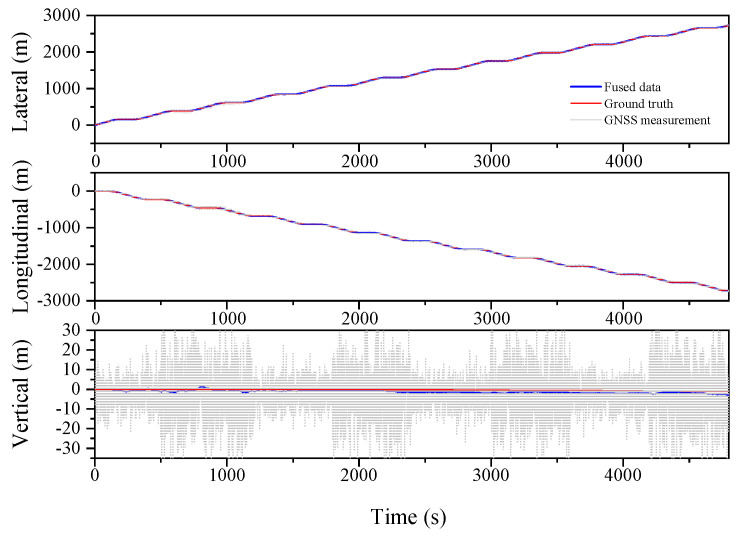
Lateral, Longitudinal and Vertical positions of ESKF (Serpentine 2).

**Figure 12 sensors-23-03676-f012:**
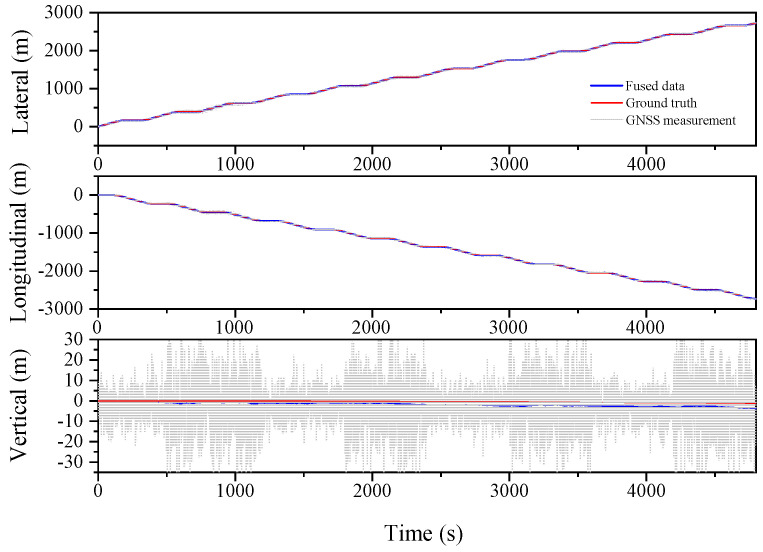
Lateral, Longitudinal and Vertical positions of ESKF−RTS (Serpentine 2).

**Figure 13 sensors-23-03676-f013:**
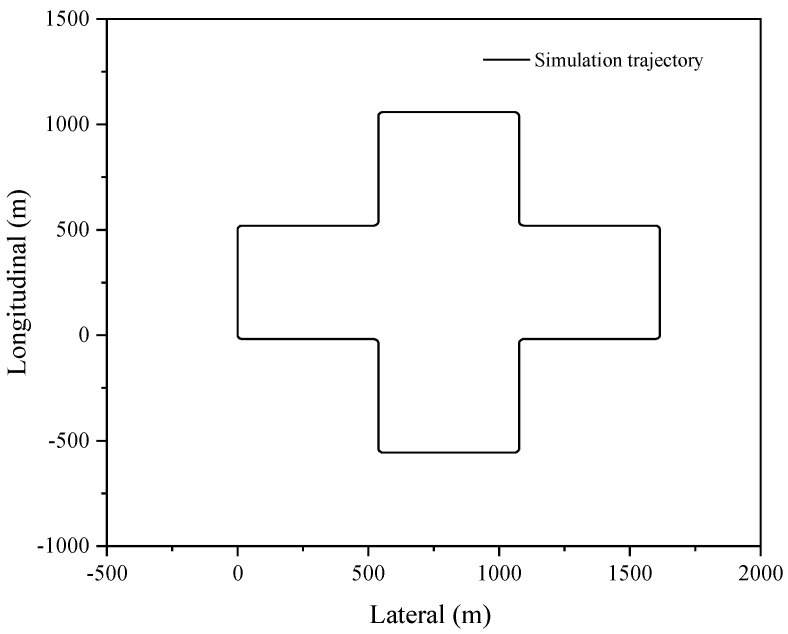
Motion trajectory of polygon.

**Figure 14 sensors-23-03676-f014:**
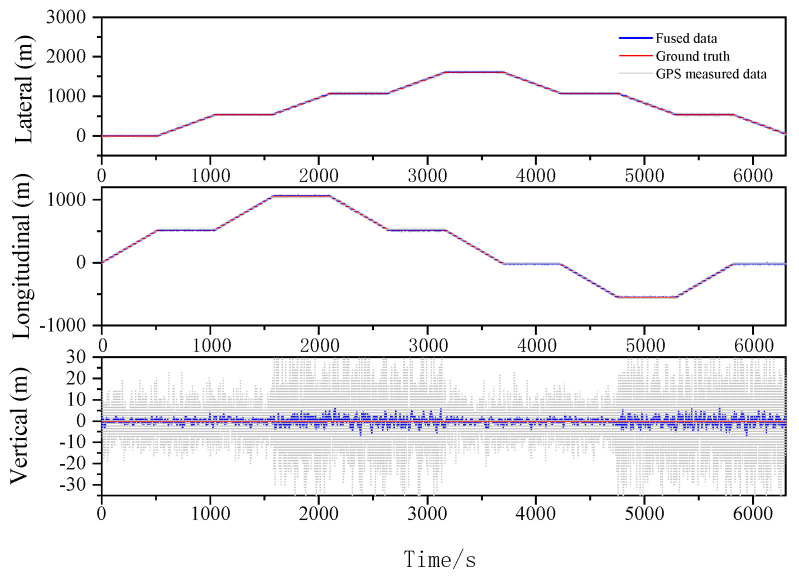
Lateral, Longitudinal and Vertical positions of EKF (Polygonal).

**Figure 15 sensors-23-03676-f015:**
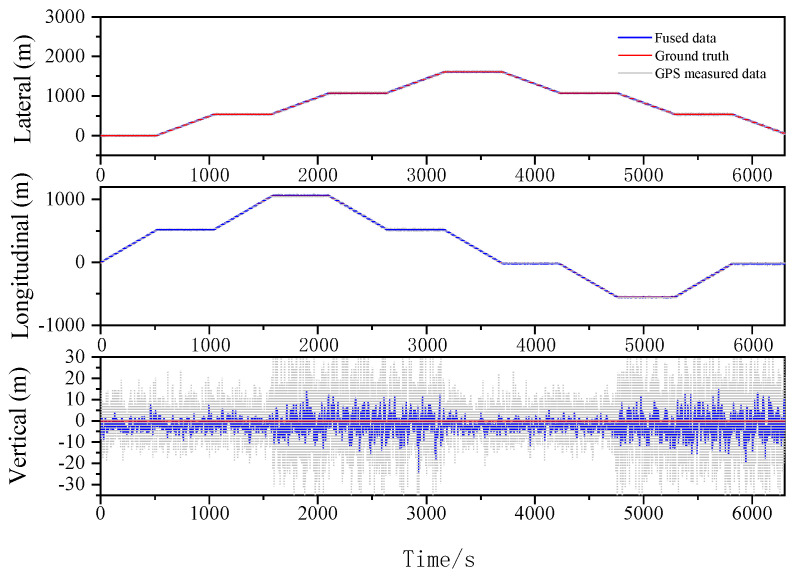
Lateral, Longitudinal and Vertical positions of ESKF (Polygonal).

**Figure 16 sensors-23-03676-f016:**
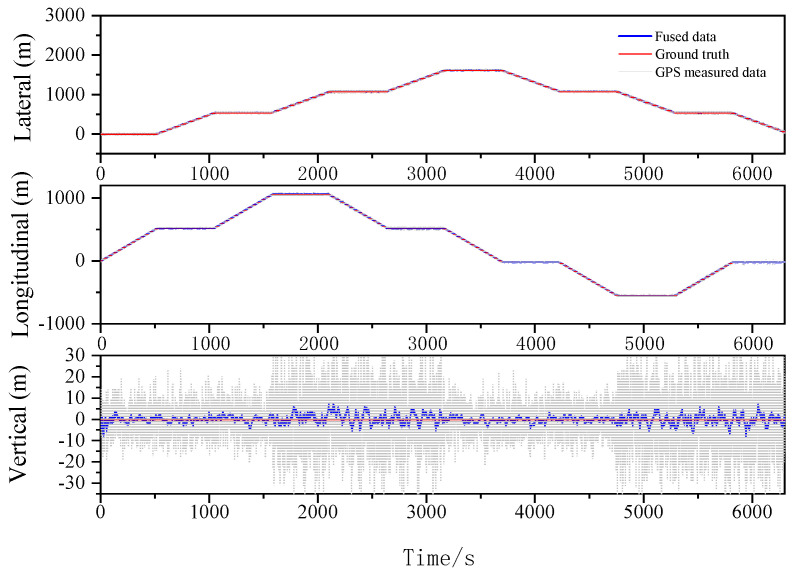
Lateral, Longitudinal and Vertical positions of ESKF−RTS (Polygonal).

**Table 1 sensors-23-03676-t001:** The definition of variable symbols.

Name	True State	Nominal State	Error States
All states	$x_{t}$	x	δx
Location	$p_{t}$	p	δp
Speed	$v_{t}$	v	δv
Quaternion	$q_{t}$	q	δq
Rotation matrix	$R_{t}$	R	δR
Angular vectors	$\theta_{t}$	θ	δθ
Acceleration bias	$a_{bt}$	$a_{b}$	$\delta a_{b}$
Angular velocity bias	$\omega_{bt}$	$\omega_{b}$	$\delta \omega_{b}$

**Table 2 sensors-23-03676-t002:** RMS values of localization errors for different algorithms (Oval).

Position Error RMS (m)	Lateral	Longitudinal	Vertical
GNSS	2.049	2.598	2.619
EKF	1.680	1.820	3.075
ESKF	1.633	1.782	1.476
ESKF–RTS	1.463	1.588	1.393

**Table 3 sensors-23-03676-t003:** RMS values of localization errors for different algorithms (Serpentine 1).

Position Error RMS (m)	Lateral	Longitudinal	Vertical
GNSS	1.912	1.793	5.680
EKF	0.955	1.585	5.823
ESKF	0.464	0.641	1.700
ESKF−RTS	0.206	0.243	0.912

**Table 4 sensors-23-03676-t004:** RMS values of localization errors for different algorithms (Serpentine 2).

Position Error RMS (m)	Lateral	Longitudinal	Vertical
GNSS	3.370	3.308	8.988
EKF	1.231	1.735	1.453
ESKF	0.635	0.890	0.957
ESKF−RTS	0.368	0.422	1.456

**Table 5 sensors-23-03676-t005:** RMS values of localization errors for different algorithms (Polygonal).

Position Error RMS (m)	Lateral	Longitudinal	Vertical
GNSS	2.601	2.677	8.516
EKF	1.243	1.289	4.328
ESKF	1.253	1.320	1.237
ESKF−RTS	0.900	0.896	1.999

## Data Availability

The data that support the findings of this study are available from the corresponding author, Xiaobin Ning, upon reasonable request.

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
