# Peer review of "Sensor Fusion of GNSS and IMU Data for Robust Localization via Smoothed Error State Kalman Filter"

_sensors, 2023, doi:10.3390/s23073676_

Round 1
Reviewer 1 Report
The paper addresses an interesting issue, that is fusion of GNSS and INS data to improve positioning accuracy.
Although the authors mostly focus on the adoption of Kalman filtering, several other approaches were proposed in the literature to address this issue, see for example:
M. M. Atia et al., "A Low-Cost Lane-Determination System Using GNSS/IMU Fusion and HMM-Based Multistage Map Matching," in IEEE Transactions on Intelligent Transportation Systems, vol. 18, no. 11, pp. 3027-3037, Nov. 2017, doi: 10.1109/TITS.2017.2672541.
S. S. Kourabbaslou, A. Zhang and M. M. Atia, "A Novel Design Framework for Tightly Coupled IMU/GNSS Sensor Fusion Using Inverse-Kinematics, Symbolic Engines, and Genetic Algorithms," in IEEE Sensors Journal, vol. 19, no. 23, pp. 11424-11436, 1 Dec.1, 2019, doi: 10.1109/JSEN.2019.2935324.
A wider analysis of the state of the art would thus improve the amnuscript.
The authors propose a Error-State Kalman Filter (ESKF) to perform the fusion, combined with data smoothing. As it is, it seems that the main innovation of the paper is the introduction of smoothing, since ESKF was previously proposed from the literature. A clear definition of the paper contributions would halp to asses its novelty and technical significance.
The description of the algorithm is very cumbersome and hard to follow: although an analytical definition of the different steps of the algorithm is definitely needed, the description could and should be revised in order to improve readability and to better explain the role of each step.
The performance evaluation is rather limited, as it is based on two synthetic trajectories and simulated data. Given the availability of hardware for both considered technologies, the authors should consider an experimental validation of the proposed algorithm
Finally, the quality of presentation should be markedly improved, in particular in terms of grammar and syntax.
Reviewer 2 Report
This paper presented sensor fusion of gnss and imu for localization. Overall, the structure of this paper is well organized, and the presentation is relatively clear. The idea is interesting and has potential. However, there are still some problems that need to be carefully addressed before a possible publication. More specifically,
1. As you have mentioned, GNSS+IMU localization is important for intelligent vehicles. Some related work should be discussed. For example, it could provide the position, velocity and attitude information for vehicle control: Vision-aided intelligent vehicle sideslip angle estimation based on a dynamic model; Automated vehicle sideslip angle estimation considering signal measurement characteristic; IMU-based automated vehicle body sideslip angle and attitude estimation aided by GNSS using parallel adaptive Kalman filters.
2. Some high-quality GNSS+IMU work for vehicle navigation should be discussed in depth in the introduction: autonomous vehicle kinematics and dynamics synthesis for sideslip angle estimation based on consensus kalman filter; improved vehicle localization using on-board sensors and vehicle lateral velocity; estimation on imu yaw misalignment by fusing information of automotive onboard sensors.
3. The Kalman filter variants with related references should also be discussed in the introduction to highlight the advantage of your adopted Kalman filter method for vehicle navigation applications.
4. Future work should be discussed at the end of the paper.
Reviewer 3 Report
Attached
